# Antibacterial and Osteogenic Activity of Titania Nanotubes Modified with Electrospray-Deposited Tetracycline Nanoparticles

**DOI:** 10.3390/nano10061093

**Published:** 2020-06-01

**Authors:** Su-Yeon Im, Kwang-Mahn Kim, Jae-Sung Kwon

**Affiliations:** 1Department of Convergent Health Science and 3D Printing, Dongnam Health University, Suwon 16328, Korea; syidt@dongnam.ac.kr; 2Department and Research Institute of Dental Biomaterials and Bioengineering, Yonsei University College of Dentistry, Seoul 03722, Korea; kmkim@yuhs.ac; 3BK21 PLUS Project, Yonsei University College of Dentistry, Seoul 03722, Korea

**Keywords:** titanium, titania nanotubes, implants, tetracycline, electrospray deposition, antibacterial, osteogenic activity

## Abstract

The nanotubular surface of titanium implants is known to have superior osteogenic activity but is also vulnerable to failure because of induced bacterial attachment and consequent secondary infection. Here, the problem was attempted to be solved by depositing nanosized tetracycline (TC)-loaded particles in poly(lactic-co-glycolic acid) on titania nanotubes (TNTs) using the electrospray deposition method. The antibacterial effect of the newly formed TNT surface was considered using the common pathogen *Staphylococcus aureus*. Maintenance of the biocompatibility and osteogenic characteristics of TNTs has been tested through cytotoxicity tests and osteogenic gene expression/extra-cellular matrix mineralization, respectively. The results showed that TNTs were successfully formed by anodization, and the characterization of TC deposited on the TNTs was controlled by varying the spraying parameters such as particle size and coating time. The TC nanoparticle-coated TNTs showed antibacterial activity against *Staphylococcus aureus* and biocompatibility with MC3T3-E1 pre-osteoblasts, while the osteogenic activity of the TNT structure was preserved, as demonstrated by osteocalcin and osteopontin gene expression, as well as Alizarin red staining. Hence, this study concluded that the electrosprayed TC coating of TNTs is a simple and effective method for the formation of bactericidal implants that can maintain osteogenic activity.

## 1. Introduction

Successful restoration using bone implants depends on secure bonding between the biomaterial and bone tissue [1]. The surface properties of a biomaterial are critical factors that play a decisive role in the success of implant surgery and allow the induction of early adhesion and differentiation of osteogenic cells to ultimately achieve bone regeneration [2,3].

The oxide film on the surface of titanium (Ti) has excellent biocompatibility [4] and the ability to integrate with bone by a process known as osseointegration [5]. Thus, it is the most widely used biomaterial in implants. As a result that the oxide film of a Ti surface is only approximately 100 Å thick [5], various surface treatments, including resorbable blasted media (RBM), sand-blasted large-grit acid-etching (SLA) and anodization of titania nanotubes (TNTs), have been developed to thicken the film and endow it with topographic characteristics that would allow it to have more bone-to-implant contact [6,7,8]. Among these surface treatments, anodization of TNTs has the most tuneability to permit deliberate adjustment of the diameter and length of the tubes [9,10]. Therefore, this treatment possesses the potential to induce quick osseointegration and excellent implant fixation at an early phase by controlling the stem cell fate to adhere, migrate or differentiate [11,12,13].

Despite these advantages of TNTs, it has also been suggested that such treatment on implants is vulnerable to failure because of the increased chance of bacterial attachment compared to conventional machined Ti surfaces [14]. To prevent bacterial infections at the site of the implant, systemic delivery of antibiotics has been typically prescribed after surgery; however, these drugs are accompanied by possible antibiotic-associated side effects on other systemic organs [15]. Therefore, the investigation of antibacterial coatings has been progressing to meet the need for an implant surface with both satisfactory localized osteogenesis and antibacterial activity.

Tetracycline (TC) is a broad-spectrum antibiotic that inhibits bacterial protein synthesis, periodontitis and osteomyelitis [16,17,18]. Biodegradable polymers such as poly(lactic-co-glycolic acid) (PLGA) are regularly used as TC carriers due to improved drug encapsulation, controlled release and reduced cytotoxicity compared to that of other materials [19]. Many of the currently available antibacterial coating methods, such as the dipping method, result in excellent long-term release [20] but lose the benefit of the nanotubular structure as the antibacterial coating covers the TNT surface. To compensate for this shortcoming, the nanoparticle coating process can be conducted by emulsification-diffusion [21], solvent-emulsion evaporation [22], magnetron sputtering [23] or electrospray deposition (ESD) [24].

ESD involves spraying a solvent that has been atomized by applying high-voltage electricity to the substrate. The major advantage of ESD is the precise control of the thin films through adjustment of the solvent type, deposition rate, spray distance, nozzle diameter, etc. Additionally, compared to other vapor deposition methods, ESD is conducted in an environment at room temperature and atmospheric pressure; thus, the fabrication process has the advantage of being uncomplicated and cost-effective [25,26,27].

Surface attachment of cells and bacteria occurs simultaneously and competitively [28,29], therefore this study attempted to produce a surface on which the TNTs are nontoxic to cells but destroy bacteria. In this in vitro study, we investigate a TNT surface modified by anodization with time-dependent electrosprayed TC nanoparticles on which adhesion of *Staphylococcus aureus* (*S. aureus*) is prevented while osteogenic activity of MC3T3-E1 is maintained due to TNT topographical effects.

## 2. Materials and Methods

### 2.1. TNT Fabrication

Ti discs (1.2 mm in diameter and 1 mm in thickness, 99.99% purity; Hyundai Titanium, Incheon, Korea) were polished with a 3 µm diamond abrasive paste on a polishing cloth and ultra-sonically degreased in acetone, ethanol and deionized (DI) water, successively, for 5 min each. TNTs were fabricated using an electrochemical anodization method at 20 V for 1 h using a platinum plate as the cathode in an electrolyte that consisted of 0.5 wt.% hydrofluoric acid [10]. After anodization, the specimens were rinsed with DI water and dried in a vacuum chamber for 1 d. Heat treatment was performed at 450 °C for 3 h in a furnace. All specimens were sterilized by ethylene oxide (EO) gas before the in vitro tests.

### 2.2. Drug Loading with ESD

TC (0.1 wt.%; Sigma-Aldrich, St. Louis, MO, USA) and PLGA (0.1 wt.%, 50:50, mol wt. 40,000–75,000; Sigma-Aldrich) were dissolved in dichloromethane (Sigma-Aldrich) and stirred with an agitator for 12 h. The solution was then centrifuged at 4500 rpm for 15 min, and the supernatant was extracted. Drug loading was performed using an ESD device (NNC-ESP200R, NanoNC, Seoul, Korea) at a constant voltage of 25 kV. A disposable syringe was filled with TC-PLGA liquid, and the flow rate was set to 10 µL/min using an infusion pump (KSD 100, KD Scientific, Holliston, MA, USA). The distance between the precision nozzle and the TNT substrate was set to 15 cm, and the deposition time was varied from 2 to 60 min (Figure 1). Simple machined Ti (as supplied by the manufacturer before TNT fabrication) and TNT substrate without ESD deposition were used as the controls. All the control and test groups are summarized in Table 1 with an assigned code for the purpose of this study.

### 2.3. Surface and Particle Characterization

X-ray diffraction (XRD; Ultima IV, Rigaku, Tokyo, Japan) was conducted on the T0 sample before and after the heat treatment as well as on the MA sample to detect and compare the crystallization of TNT to simply machined Ti. Transmission electron microscopy (TEM; JEM-2010, JEOL, Tokyo, Japan) and dynamic light scattering (DLS; ZEN 3690, Malvern Instruments, Malvern, UK) were used to confirm the shape and diameter of TC particles. The particles for DLS were suspended in water with electrospray deposition. Field-emission scanning electron microscopy (FE-SEM; JSM-6701F, JEOL, Tokyo, Japan) was used to analyze the structure of the TNTs as well as the morphology of the deposited TC particles on the test substrates.

### 2.4. Antibacterial Assay

*S. aureus* (American Type Culture Collection (ATCC) 29213, Manassas, VA, USA) was cultured aerobically in brain heart infusion (BHI; Becton, Dickinson and Company, Franklin Lakes, NJ, USA) medium. The bacteria were seeded on substrates at an initial density of 1.0 × 10^7^ colony forming units (CFU)/mL and incubated at 37 °C for 24 h.

Antibacterial activity was investigated using the SYTO9 and propidium iodide (PI) LIVE/DEAD BacLight Bacterial Viability/Cytotoxicity Kit™ (Invitrogen, Carlsbad, CA, USA). The bacteria on the functionalized substrates were gently washed with phosphate-buffered saline (PBS; Gibco, Carlsbad, CA, USA) and stained with a mixture of two-color nucleic acid stains for 15 min. SYTO9 stains normal bacteria fluorescent green according to the condition of the bacterial membrane, while dead bacteria are stained fluorescent red by PI. Immunofluorescence imaging was performed by confocal laser scanning microscopy (CLSM; LSM700, Carl-Zeiss, Oberkcochen, Germany).

For the quantification of bacterial attachment, bacteria were cultured for 1 d and rinsed with PBS followed by placement into a glass vial containing new BHI medium. The samples were ultrasonicated for 5 min to detach the bacteria on the surface of the specimen. The extracted bacterial suspension was diluted 10-fold in series, spread on BHI agar plates and then incubated at 37 °C for 1 d. Afterwards, evaluation of antibacterial activity was performed by counting the number of visible *S. aureus* CFU.

### 2.5. Cell Culture

Mouse pre-osteoblasts (MC3T3-E1, subclone 4; ATCC) were cultured in alpha minimum essential medium (Invitrogen) supplemented with 10% fetal bovine serum (Invitrogen) and 1% antibiotics/antimycotics (Invitrogen). The cells were incubated in a humidified atmosphere with 5% CO_2_ at 37 °C until the experiments were performed, and cells with passages less than 5 were used for the experiments. The cells were placed on the samples at an initial density of 1.0 × 10^5^ cells/mL.

### 2.6. Cytotoxicity Assay

Cytotoxicity was assessed with calcein AM and ethidium homodimer-1 (ethD-1) staining (LIVE/DEAD Viability/Cytotoxicity Kit™, Invitrogen) assays. After culturing cells on each control and test sample for 1 d, the cells on the substrates were washed in Dulbecco’s phosphate-buffered saline (DPBS; Invitrogen) and stained with calcein AM for live cells (green) and ethD-1 for dead cells (red). The images of stained cells were observed using a confocal laser microscope (LSM700, Carl-Zeiss).

### 2.7. Cell Morphology

To evaluate the morphology of attached cells, the cells were cultured on each control or test sample for 24 h and then washed with wash buffer (0.05% Tween 20 in PBS), followed by fixation with 4% paraformaldehyde. Fixed cells were then stained for 1 h with fluorescein isothiocyanate (FITC)-labeled vinculin (Millipore, Billerica, MA, USA), which indicates focal adhesions (green) and Tetramethylrhodamine (TRITC)-conjugated phalloidin, which indicates actin filaments (red). The morphology of the cells was observed using CLSM (LSM700, Carl-Zeiss).

### 2.8. Osteogenic Gene Expression

The expression levels of the osteogenic markers osteopontin (OPN) and osteocalcin (OCN) were detected by real-time reverse transcription-polymerase chain reaction (real-time RT-PCR). Briefly, after 21 d of culturing the cells on each control or test sample, RNA from the cells was extracted with TRIzol (Invitrogen). Total RNA was reverse transcribed to complementary DNA (cDNA) using a high-capacity RNA-to-cDNA kit (Applied Biosystems, Carlsbad, CA, USA). For DNA amplification, solutions with specific primers and SYBR green (Applied Biosystems) were added to the respective cDNA samples. Real-time PCR was then performed using an ABI Prism 7500 machine (Applied Biosystems). The expression level of each osteogenic gene was normalized against the amount of glyceraldehyde 3-phosphate dehydrogenase. To confirm the findings, immunofluorescence staining was performed with solutions of anti-OPN antibody (Santa Cruz Biotechnology, Dallas, TX, USA) and anti-OCN antibody (QED Bioscience, San Diego, CA, USA) in blocking solution (1% BSA in PBS) at 4 °C for 1 d. After the cells were rinsed with wash buffer, they were soaked in solution with goat anti-mouse lgG-FITC (Santa Cruz Biotechnology, Dallas, TX, USA) for 1 h. The stained cells were observed using CLSM (LSM700, Carl-Zeiss).

### 2.9. Extracellular Matrix Mineralization

Extracellular matrix mineralization was assessed using Alizarin red S (ARS; Sigma-Aldrich) staining method. After culturing cells on each control or test sample for 3 weeks, adherent cells were washed with PBS, fixed with absolute ethanol and stained using 2% ARS solution for 15 min. Stained samples were then washed with DI water, and red-colored extracellular matrix mineralization was photographed with an optical microscope. For the colorimetric assay, the specimen was soaked in cetylpyridinium chloride (CPC; Sigma-Aldrich) solution for 1 h so that the stain on the specimen would dissolve. The absorbance of the CPC solution was then measured at 560 nm using a spectrophotometer (Epoch, BioTek Instruments, Winooski, VT, USA).

### 2.10. Statistical Analysis

All experiments were repeated in triplicate. Quantitative data were expressed as the mean ± standard deviation and analyzed with SPSS 20.0 software (SPSS Inc., Chicago, IL, USA). One-way ANOVA combined with Tukey’s post hoc test was used to determine the level of significance. A value of *p* < 0.05 was considered to be significant.

## 3. Results

### 3.1. Surface Characterization

The XRD spectra were obtained from T0 before and after heat treatment as well as from the MA groups to characterize their surface crystallinity (Figure 2a). It was noted that the (1 0 1) anatase diffraction peak was only identified in heat-treated T0 (T0-AH in Figure 2a) [30].

The diameter of PLGA-coated TC particles produced by the ESD method was approximately 24.36 nm, as shown in Figure 2b, which was confirmed along with the morphology of the particles using TEM and DLS (Figure 2c).

The morphology of the control group and the ESD-sprayed test group was verified by FE-SEM (Figure 2d). The surface of the MA group made by grinding pure Ti was flat, whereas the surface of T0 prepared by anodization treatment with MA as a substrate presented TNTs with regularly shaped nanotubes that were 100 nm in diameter. The structure of the tubes was shown to be preserved even after heat treatment. For the samples deposited with TC particles by ESD with increasing spraying time, the applied particles were evenly distributed along the tube structures, maintaining the unique TNT topography until 16 min of deposition (T2, T4, T8 and T16 in Figure 2d). However, it was observed that the entrances of the TNTs were completely sealed following 30 min of deposition (T30 in Figure 2d), whereas the tube-like structures were difficult to be seen in the FE-SEM image following 60 min of deposition (T60 in Figure 2d) due to the TC layer.

### 3.2. Antibacterial Activity

Figure 3a shows the immunofluorescence following the culture of *S. aureus* on each specimen. The number of intact live bacteria, which exhibited a green color, was greater in the T0 group than in the MA group. However, the number of viable *S. aureus* decreased with increasing spraying time. The number of viable bacteria was significantly reduced in groups T2 and T4 compared with both MA and T0, with an increased number of dead bacteria (red color). Furthermore, no intact bacteria were observed in groups T8, T16, T30 and T60.

Additionally, residual bacterial proliferation was confirmed with the CFU count method (Figure 3b). The number of colonies was approximately 240 CFU/mL in MA, whereas the T0 group showed CFU counts of approximately 320 CFU/mL, which is 33% higher than that in the MA group. However, the CFU numbers decreased significantly to 101 and 25 in the T2 and T4 groups, respectively, both of which received antibiotics. No colonies were observed in the T8 group or in any of the other groups with a longer duration of deposition than that of T8, proving the antimicrobial characteristic of TC.

### 3.3. Cytotoxicity

As stated above, an overdose of TC may be toxic to cells. Therefore, the cytotoxicity was evaluated, and the results are shown in Figure 4. The patterns of cell distributions and the viability of cells in groups T2, T4 and T8 were similar to those in T0 despite the increase in spraying time. However, when the spraying time reached 16 min (T16), the cells became thinner, and dead cells that were stained red started to appear (arrows on Figure 4). Furthermore, most cells in the T30 groups had a round shape with an increasing number of dead cells, while most of the cells in T60 appeared to be circular and dead.

### 3.4. Cell Morphology

The morphology of attached cells was observed by confocal laser microscopy to investigate the effects of surface treatment on the cells (Figure 5). In the T0 group, the cytoskeleton was well developed relative to the MA group, with extended actin filaments (red) and focal adhesions as shown by vinculin (green). This morphological shape was relatively unchanged in the T8 group in which the TC particles were introduced.

### 3.5. Gene Eexpression

The gene expression of OCN and OPN, markers of osteogenesis, was assessed by both immunofluorescence and RT-PCR to determine the bioactivity of the control and test specimens (Figure 6). The RT-PCR results showed the upregulation of osteogenic differentiation by osteoblast cells on the nanotubular surface (T0) relative to the flat surface (MA) (Figure 6a). This effect was maintained even after the application of TC on the surface of the nanotubes using the ESD method (T8). The result was then confirmed with immunofluorescence, which also showed upregulation of OCN in T0 relative to the MA group (Figure 6b). Additionally, OPN was significantly more densely expressed in the former group, exhibiting the same pattern as the cells. There was no significant difference in the amounts of OCN and OPN expression between the T8 and T0 groups; both groups had relatively high levels of expression.

### 3.6. Extracellular Matrix Mineralization

Extracellular matrix mineralization was measured with ARS staining (Figure 7a). The surface of the MA group showed a weak distribution of crystals, while the T0 group had increased calcium content, which was evident as reddish-black coloration and wave-shaped crystallization nodules (arrows in Figure 7a). In addition, the calcium crystallization rate in the T8 group was similar to that in the T0 group. Quantitative analysis through measurement of the absorbance showed that the absorbance value was significantly higher by 23% in the T0 group than in the MA group, while there was no statistically significant difference in absorbance between groups T0 and T8 (Figure 7b).

## 4. Discussion

Infection is the main reason for early-stage implant failure due to its interference with the bonding of the implant surface with bone tissue [28,31]. Therefore, the development of an implant surface that possesses both osteogenic abilities and antibacterial activity is our ultimate long-term objective [32].

To develop a bioactive surface, many studies have applied growth factors to improve the osteogenic abilities [33,34] and either antibiotics [15] or inorganic nanoparticles [23] to impart the implant surface with antibacterial properties. Despite the excellent results reported from these studies, there are clearly also challenges [35]. Even if a growth factor possesses excellent osteogenic abilities, high cost and side effects have often been reported [36], whereas inorganic nanoparticles that are effective against antibiotic-resistant bacteria need verification of their long-term cytotoxicity as they are not degraded in vivo [23,37]. Additionally, the dipping method of antibiotic polymer coatings for long-term drug release [20] has been known for the difficulty of forming a uniform coating layer and inability to conserve the surface structure of the substrate.

Hence, to overcome these problems, this study aimed to apply and verify the ESD method for nontoxic antibacterial activity by forming a compact TC nanoparticle pattern while preserving the osteogenic ability of TNTs. The ESD method has been applied in various biomedical areas, including implant coating [38], scaffold production [39] and biosensors [40]. By adjusting conditions such as the voltage, the distance between the nozzle and the substrate, the applied pressure on the solvent and the duration of spraying, ESD can easily be used to form a surface with the preferred layer characteristics, as observed with the test samples of this study (Figure 1) [26]. In a previous study where the particles created by the ESD method resulted in an average diameter of 100 nm [41], our method of using a simple centrifuge stage resulted in PLGA-coated TC particles with an average diameter of 24.36 nm (Figure 2c). Smaller particle sizes have many advantages for ESD applications, such as the prevention of metal nozzle blockage, which is a common problem that can occur when using ESD equipment. In addition, because the sprayed droplets in the ESD method not only attain a mono-disperse distribution but are also charged, they demonstrate the merit of decreasing the tendency for droplets to aggregate with each other [26]. Thus, particles were evenly sprayed along the structure of the tube with a diameter of 100 nm, as seen from T2 to T16 in Figure 2d. However, when the spraying time increased to T30, there was an intensified aggregation of particles that resulted in the formation of a layer-like structure, which completely closed the tubular structures. The thickness of the antibacterial complex layer is expected to determine the bactericidal effectiveness and toxicity to the cell [42]. Additionally, covering the nanotubular structures, which contribute to the osteogenic properties, may result in a loss of the bioactive features of the surface [12]. Therefore, the rest of the investigations in this study focused on seeking the critical point at which the antibacterial activity and osteogenic properties of the nanotubular structures were maintained without causing cytotoxicity.

Following heat treatment, the TNTs had a low contact angle, which indicates that high surface energy contributes to the blood infiltrating the surface and enhancing osteoblast attachment. Additionally, the development of the anatase phase, as confirmed by XRD following heat treatment of the TNTs, is expected to contribute to the favorable osteogenic properties of the Ti surface (Figure 2a) [11,43,44].

The antibacterial activity of TC nanoparticles on specimens has been demonstrated with different degrees of bacterial attachment (Figure 3a). Initially, there was an increased number of *S. aureus* cells attached to T0 compared to the flat MA surface, as the surface provided a more favorable surface for the bacteria to attach as the surface area increased [14]. Rapid decreases in the number of living bacteria were observed from T2 to T8 as the loading and release of TC gradually increased. PI infiltrates bacteria possessing damaged membranes and stains the nucleic acids red [45]. However, fluorescent red bacteria were not observed from T8 to T16 because the damaged bacteria on TC were no longer attached to the substrate but instead were washed away during the staining process. These results were then confirmed by the results shown in Figure 3b, where CFU counts were used to monitor the residual bacterial growth. There were statistically significant decreases in the number of bacteria in T2 and T4, and bacterial colonization was suppressed from T8 onward.

Despite the positive antibacterial effects of TC shown above, release of TC nanoparticles from the biomaterial needs to be controlled because they can induce cytotoxicity at overdose levels [42]. The cytotoxicity of each test group was confirmed through the assessment of cell viability using the LIVE/DEAD assay (Figure 4). The results showed that each cell had a dramatic radial spread in T0 compared to that in MA. The same tendency has been observed from T0 to T8, therefore it seems to be an effect of the TNT structure [11,12,13]. Additionally, there was no significant difference in the number of attached cells, but we found that the cells had already started to transform from T16 and that dead cells began appearing, as confirmed through CLSM (white arrows). The number of undamaged cells decreased in T30, and dead cells that were completely round and destroyed by the toxicity of TC were observed in T60. The ultimate objective of processing the implant surface was to induce osseointegration [1], therefore we selected 1 group from the test group, T8, which was shown to maintain the TNT structure without being covered by the TC layer and to impede colonization of bacteria without causing cytotoxicity to host cells. This group was compared with MA and T0 to investigate the maintenance of cell attachment and the osteogenic differentiation potency of the TNTs even after ESD spraying with TC. The results showed that the morphology of cell attachment on T8 was similar to that on T0 because the topographical cue had been preserved (Figure 5). The TNT structure contains an empty space in the middle that gives a long distance between the parts that the cells can attach to, which manifests as an elongation of actin filaments and the cytoskeleton. Although some particles that have been sprayed at the edge of the tube exist in a random nanonodule shape (Figure 2c), there was no significant effect on focal adhesion formation [46,47,48].

Finally, the ability of TNTs to induce osteogenic differentiation without any chemical factors was investigated. The RT-PCR results showed that the gene expression of the protein osteogenic markers OCN and OPN was upregulated and induced differentiation into osteoblasts in both T0 and T8 compared with the non-TNT surface of the MA group. The high number of fluorescent signals in OCN and OPN present in T0 and T8 also confirmed that osteogenic signaling was initiated due to the applied tension to cells through stretching by the TNT surface structure [13]. Furthermore, the results of ARS staining showed that there was no significant difference in nodule formation and extracellular matrix mineralization between T0 and T8 (Figure 7). These results clearly suggest that TC particles sprayed on the T8 surface using the ESD method do not hinder osteoblast differentiation compared with unsprayed TNT surfaces.

This study showed that a TC-sprayed surface such as T8 formed using ESD demonstrated antibacterial activity and osteogenic abilities. The ESD method can provide antibacterial activity on the surface without causing cytotoxicity, while the properties of the TNT structure were preserved from such an antibacterial coating in terms of the stimulation of OCN and OPN signaling, thus inducing osteogenic differentiation. However, there is limited prevention of secondary infections after the complete fixation of implants because only a minimal quantity of TC has been sprayed to preserve the TNT surface structure. In addition, the wear resistance of the surface during implantation is one of the tasks to be overcome. Nonetheless, we can expect antibacterial effects against the invasion of foreign bacteria while the osteogenic ability is maintained during the early critical period, which is when the cells attach onto the surface and differentiate. Our approach also has the potential to be used to complexly apply various other ingredients on top of nanotubes using a similar process, which is a topic of further research.

## 5. Conclusions

To improve the antibacterial and osteogenic activity of the TNT surface, TC particles were deposited using the ESD method. Our results showed that the distribution along the tube structures and release of TC nanoparticles can be controlled by the spraying time. The T8 group showed significant antimicrobial activity without cytotoxicity. Moreover, the deposition of TC in the T8 group maintained the osteogenic activity of the TNTs, as demonstrated by the morphology of cells but also by inducing osteogenic effects of osteoblasts as good as those observed in the T0 group. These results show that, despite some limitations, TNTs coated with TC nanoparticles using the ESD method have osteogenic and antibacterial activity, which may be able to reduce the failure rate of implant surgery.

## Figures and Tables

**Figure 1 nanomaterials-10-01093-f001:**
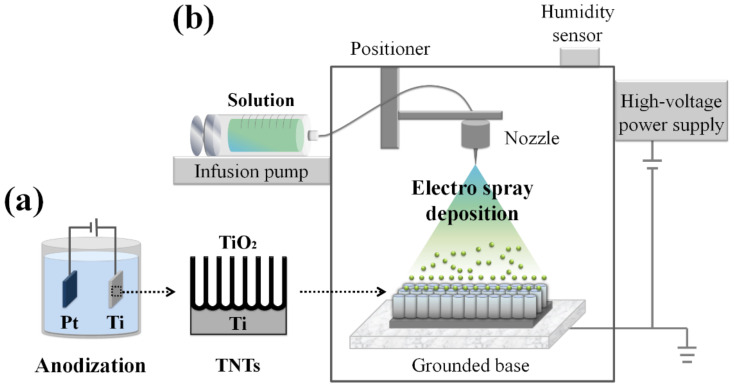
Schematic drawings of the (**a**) electrochemical anodization and (**b**) electrospray deposition system.

**Figure 2 nanomaterials-10-01093-f002:**
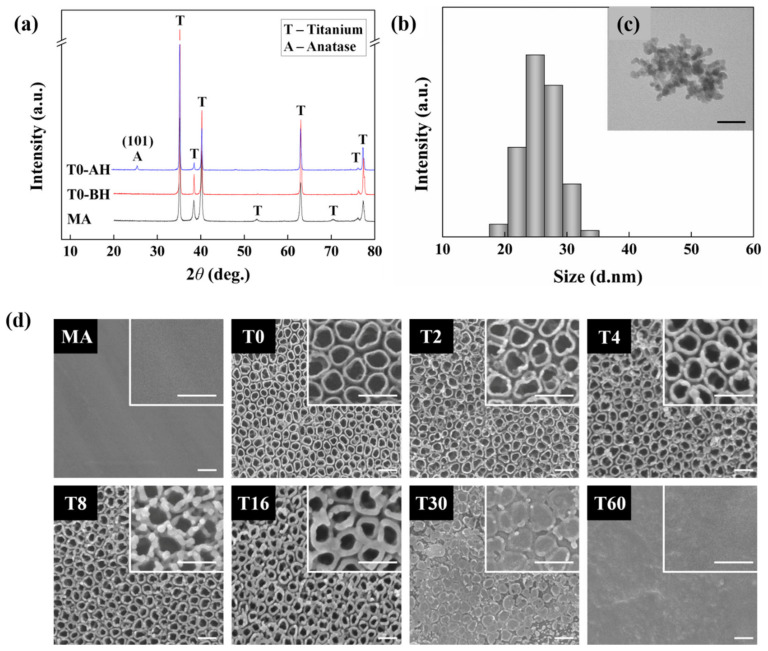
Surface characterization of control and test specimen. (**a**) XRD spectra of machined Ti (MA), non-heat-treated T0 (T0-BH) and T0 heat-treated at 450 °C (T0-AH), characterization of electrospray-deposited tetracycline nanoparticles by (**b**) dynamic light scattering for the nanoparticle size and (**c**) transmission electron microscopy for the surface morphology of the nanoparticles (scale bar = 100 nm), and (**d**) surface morphology of the specimens assessed by FE-SEM (top view) (scale bar = 200 nm).

**Figure 3 nanomaterials-10-01093-f003:**
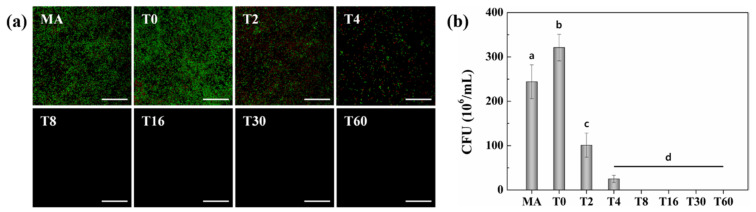
(**a**) Fluorescence images of live (green) and dead (red) stained *S. aureus* on specimens following 24 h of incubation (scale bar = 200 µm). (**b**) Antibacterial activity of specimens versus adherent *S. aureus* seeded after 24 h assessed using the CFU count method (different letters indicate significantly different groups at *p* < 0.05).

**Figure 4 nanomaterials-10-01093-f004:**
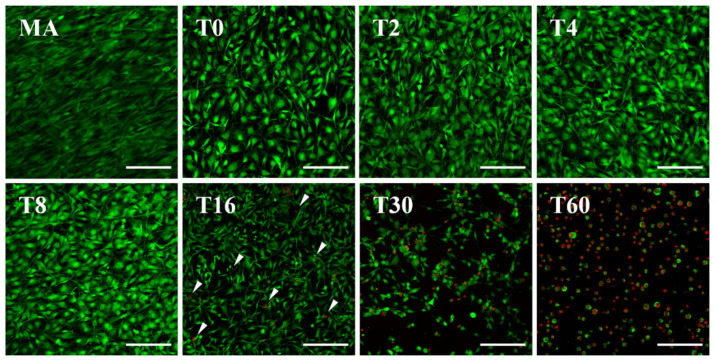
Fluorescence images for evaluation of the cytotoxicity of the surface by 24 h of cell culture using live (green) and dead (red, arrows in T16) stained osteoblasts (scale bar = 200 µm).

**Figure 5 nanomaterials-10-01093-f005:**
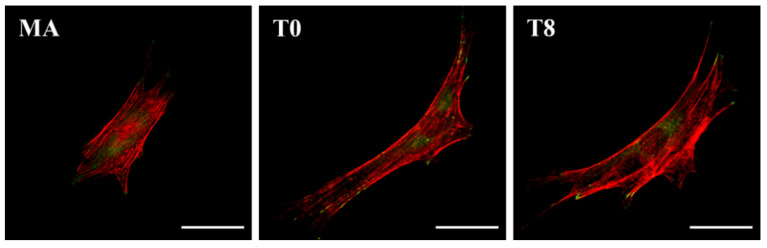
Fluorescence images of actin (red) and vinculin (green) in osteoblasts cultured on the specimens for 24 h (scale bar = 50 µm).

**Figure 6 nanomaterials-10-01093-f006:**
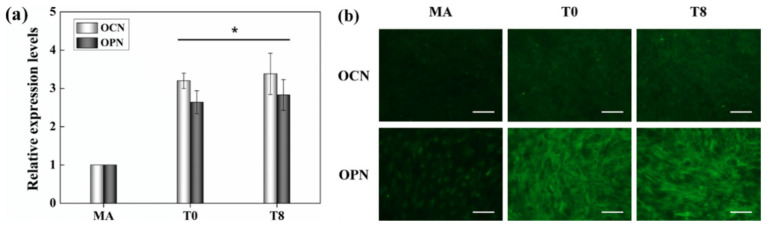
Osteogenic differentiation of osteoblasts on the specimens for 21 d. (**a**) The expression levels of osteopontin (OPN) and osteocalcin (OCN) assessed by real-time RT-PCR (**p* < 0.05 vs. MA). (**b**) Fluorescence images of OCN (upper) and OPN (lower) (scale bar = 100 µm).

**Figure 7 nanomaterials-10-01093-f007:**
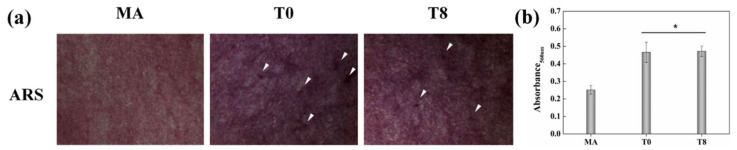
Extracellular matrix mineralization of osteoblasts on the specimens for 21 d assessed by Alizarin red S (ARS) staining. (**a**) Optical images (arrows indicate nodules) and (**b**) colorimetric quantitative analysis (**p* < 0.05 vs. MA).

**Table 1 nanomaterials-10-01093-t001:** Surface properties of substrates.

	MA	T0	T2	T4	T8	T30	T60
Surface treatment	Machined	TNT ^1^	TNT	TNT	TNT	TNT	TNT
ESD ^2^ time (min)	0	0	2	4	8	30	60

^1^ TNT: titania nanotube, ^2^ ESD: electrospray deposition.

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
