# Peer review of "Antibacterial and Osteogenic Activity of Titania Nanotubes Modified with Electrospray-Deposited Tetracycline Nanoparticles"

_nanomaterials, 2020, doi:10.3390/nano10061093_

Round 1

Reviewer 1 Report

Surface modification of titanium bone implants with antibiotic (tetracycline, TC)-loaded polymer particles. The titanium surface was first anodized to make titanium dioxide tubular structures, that were subsequently coated with the antibiotic-loaded particles by electrospray deposition. Activity against common Staphylococcus aureus was observed, as well as retained osteocompatibility of the surface.

The manuscript is well-written and easy to read. Although the work described could be potentially interesting as a proof-of-concept, it is important that the authors will clarify the following points:

  • Lines 16-17: “nanosized tetracyclin” is not correct here. “Nanosized tetracyclin-loaded particles” is more correct.
  • How practical is to perform the anodization process on actual implants? Also, how feasible is to coat these irregular surfaces conformally with ESD? Please give a comprehensive state-of-art.
  • What is the actual resistance to wear and abrasion of the resulting surfaces? Would debris be produced during the implantation, and if so, would that be a cause of inflammation?
  • For how long will the implant be actively antibiotic? How does the behavior of the sample described comply with desired and previously reported values? Please clarify and provide a comprehensive state-of-art.
  • Are there alternatives to tetracyclin? E.g. in case of allergic reactions?
  • Please give the molecular weight of PLGA used.
  • Did the authors checked the antibiotic activity of the surfaces also by the disc diffusion assay? If not, why? Please clarify.
  • How the particles for DLS measurement were collected analyzed? In which medium were dispersed, and how? Please give detailed information.
  • What is the scale bar in the insets of Figure 2d? Please clarify. Also, why the surface of sample T8 looks more particle-coated than that of sample T16?
  • Thinking in terms of real-world applications, what would be the shelf-life of the antibacterial surface? Will its activity remain the same over time under storage (under which conditions)? Did the authors considered the possible photocatalytic degradation effects of titanium dioxide, if the surfaces would be exposed to solar or UV light (e.g. for sterilization)?
  • Lines 294-297: it looks like that the centrifugation step is very important to dictate – at least – the size of the obtained particles. This raises an additional question: are the particles already formed in suspension, and then the role of ESD is only to deposit them on the surface, without influencing their size etc., or else? If the particles are already formed, please provide a mechanistic explanation for their formation and encapsulation of TC. More details are required in order to fully appreciate the particle synthesis step, as it looks like that two homogeneous solutions, both in dichloromethane, are mixed, obtaining…. something that needs to be centrifuged? To remove what? Please clarify and provide a comprehensive explanation.
  • Lines 318-319: what does it mean, “as the concentration of fluorine increased”? Is the titanium dioxide doped with fluorine as a result of the anodization? This point needs to be discussed! And evidence for fluorine doping needs to be provided.
  • The authors tried only with Staphylococcus aureus, and not even with a multi-resistant strain (MRSA), which would be much more problematic in real-world surgical settings. Also for this reason, the conclusions that could be drawn are limited (see next point).
  • Lines 375-376: the conclusion that the approach described could “reduce the failure rate of implant surgery” is a bit of a stretch, as no clinical nor in vivo results are provided. It is only possible to conclude that these results “[might be of help] to reduce the failure rate of implant surgery”.

Author Response

Reviewer #1

General Comments

Surface modification of titanium bone implants with antibiotic (tetracycline, TC)-loaded polymer particles. The titanium surface was first anodized to make titanium dioxide tubular structures, that were subsequently coated with the antibiotic-loaded particles by electrospray deposition. Activity against common Staphylococcus aureus was observed, as well as retained osteocompatibility of the surface. The manuscript is well-written and easy to read. Although the work described could be potentially interesting as a proof-of-concept, it is important that the authors will clarify the following points:

Response to General Comments

The authors would like to thank the reviewer for the interest and in-depth review with valuable and helpful comments for improvement of the manuscript.

Comment #1

(Lines 16-17)

“nanosized tetracyclin” is not correct here. “Nanosized tetracyclin-loaded particles” is more correct.

Response

Thank you for pointing this out. We have replaced the term “nanosized tetracycline” throughout the manuscript with “nanosized tetracycline-loaded particles” to use more precise terms.

Comment #2

How practical is to perform the anodization process on actual implants? Also, how feasible is to coat these irregular surfaces conformally with ESD? Please give a comprehensive state-of-art. What is the actual resistance to wear and abrasion of the resulting surfaces? Would debris be produced during the implantation, and if so, would that be a cause of inflammation?

Response

I fully agree with your opinion that there are some problems to overcome, such as the wear resistance on the implant surface and the need to rotate the actual implant during the coating method. These problems have been included and described in the Discussion section as limitations of this study. (“In addition, the wear resistance of the surface during implantation is one of the tasks to be overcome.”) In addition, we have attached the reference stating the fact that titanium debris are biocompatible as follows. (Titanium allergy: could it affect dental implant integration? (Allauddin et al. 2011); “Increased quantities of titanium ions and wear debris have been reported with implants that have a large surface area such as orthopedic implants (Clarke et al. 2003). Dental implants do not have such large surface areas, which may explain why debris has rarely been observed around failed oral implants (Esposito 2001).”)

Comment #3

For how long will the implant be actively antibiotic? How does the behavior of the sample described comply with desired and previously reported values? Please clarify and provide a comprehensive state-of-art.

Response

Thank you for providing these insights. This paper focused on whether tetracycline applied with centrifugation and the ESD coating method can destroy surface S. aureus without covering the nanotubes. Therefore, the long-term release according to the loading dosage of tetracycline has been planned as a further study.

Comment #4

Are there alternatives to tetracyclin? E.g. in case of allergic reactions?

Response

A variety of antibiotics, including indomethacin, can indeed be used with PLGA. We are planning to conduct experiments with different drugs as a further study; this future study has been mentioned in the 368th line of the Conclusions. (“Our approach also has the potential to be used to complexly apply various other ingredients on top of nanotubes using a similar process, which is a topic of further research.”)

Comment #5

Please give the molecular weight of PLGA used.

Response

The molecular weight of PLGA ranged from 40,000 to 75,000. This information has been provided on the 85th line of the Materials and Methods section. (“TC (0.1 wt%; Sigma-Aldrich, St. Louis, MO, USA) and PLGA (0.1 wt%, 50:50, mol wt 40,000-75,000; Sigma-Aldrich) were dissolved in dichloromethane (Sigma-Aldrich) and stirred with an agitator for 12 h.”)

Comment #6

Did the authors checked the antibiotic activity of the surfaces also by the disc diffusion assay? If not, why? Please clarify.

Response

I agree that disc diffusion assay is an essential way to check the antibiotic activity on the surface. To verify the antibiotic activity of the surfaces, the CFU method was mainly used in a similar prior study, and it has been supplemented by LIVE/DEAD cells assays by seeding the bacteria directly onto the surface of the specimens.

Comment #7

How the particles for DLS measurement were collected analyzed? In which medium were dispersed, and how? Please give detailed information.

Response

To provide more information and information on the medium, the distribution methods have been included in the 106th line of the Materials and Methods section. (“The particles for DLS were suspended in water with electrospray deposition.”)

Comment #8

What is the scale bar in the insets of Figure 2d? Please clarify. Also, why the surface of sample T8 looks more particle-coated than that of sample T16?

Response

Per your suggestion, we have added a scale bar to Figure 2d. If we continue to spray the nanosized tetracycline-loaded particles on top of specimen T8, it will combine and look like specimen T16. However, T16 still sees many particle shapes that can be attached to the cell.

Comment #9

Thinking in terms of real-world applications, what would be the shelf-life of the antibacterial surface? Will its activity remain the same over time under storage (under which conditions)? Did the authors considered the possible photocatalytic degradation effects of titanium dioxide, if the surfaces would be exposed to solar or UV light (e.g. for sterilization)?

Response

I fully agree with you that the shelf-life of the antibacterial surface may vary depending on factors affecting storage conditions, such as temperature and humidity. After the surface we produced was applied to the actual implant, the problem of protein deformation due to solar or UV light was designated as a part of further research.

Comment #10 (Lines 294-297)

It looks like that the centrifugation step is very important to dictate – at least – the size of the obtained particles. This raises an additional question: are the particles already formed in suspension, and then the role of ESD is only to deposit them on the surface, without influencing their size etc., or else? If the particles are already formed, please provide a mechanistic explanation for their formation and encapsulation of TC. More details are required in order to fully appreciate the particle synthesis step, as it looks like that two homogeneous solutions, both in dichloromethane, are mixed, obtaining…. something that needs to be centrifuged? To remove what? Please clarify and provide a comprehensive explanation.

Response

The key reasons for using the centrifuge and ESD methods in this paper are as follows. First, centrifugation sinks the large particles that can block the nanosized metal nozzle; then, the ESD method can be used to make the solvent evaporate naturally and spread uniformly on top of the nanotube surface to prevent particles from aggregating together. This was mentioned in the Discussion as follows. (“In a previous study where the particles created by the ESD method resulted in an average diameter of 100 nm, our method of using a simple centrifuge stage resulted in PLGA-coated TC particles with an average diameter of 24.36 nm [Fig. 2(c)]. Smaller particle sizes have many advantages for ESD applications, such as the prevention of metal nozzle blockage, which is a common problem that can occur when using ESD equipment. In addition, because the sprayed droplets in the ESD method not only attain a mono-disperse distribution but are also charged, they demonstrate the merit of decreasing the tendency for droplets to aggregate with each other.”)

Comment #11

Lines 318-319

What does it mean, “as the concentration of fluorine increased”? Is the titanium dioxide doped with fluorine as a result of the anodization? This point needs to be discussed! And evidence for fluorine doping needs to be provided.

Response

Thank you for pointing out this error; the parts mentioning fluorine were typing errors. Therefore, I have modified instances of this error as follows. (“Initially, there was an increased number of S. aureus cells attached to T0 compared to the flat MA surface, as the surface provided a more favourable surface for the bacteria to attach as the surface area increased [14].”)

Comment #12

(Lines 318-319)

The authors tried only with Staphylococcus aureus, and not even with a multi-resistant strain (MRSA), which would be much more problematic in real-world surgical settings. Also for this reason, the conclusions that could be drawn are limited (see next point).

Response

Thank you for your valuable comment. I agree it is a limitation that we experimented only with the one bacterial species of S. aureus. In further studies, we will experiment with a more diverse range of bacterial species.

Comment #13

(Lines 375-376)

The conclusion that the approach described could “reduce the failure rate of implant surgery” is a bit of a stretch, as no clinical nor in vivo results are provided. It is only possible to conclude that these results “[might be of help] to reduce the failure rate of implant surgery”.

Response

Thank you for your constructive comment. I agree with your suggestion, and I have revised the sentence in the Conclusions as follows. (“These results show that, despite some limitations, TNTs coated with TC nanoparticles using the ESD method have osteogenic and antibacterial activity, which may be able to reduce the failure rate of implant surgery.”)

Reviewer 2 Report

This is an extremely well-written manuscript detailing a procedure for depositing the anti-bacterial compound tetracycline onto titania nanotube (TNT) surfaces. Anodized TNT dental implants are used because of their ability to be osseointegrated, yet there has remained some concern in the field because of the potential for bacterial adhesion to the surface and hence bacterial infection at the implant site.  Tetracycline in PLGA is deposited by electrospray for various lengths of time to the TNT. The authors demonstrate that they produce smaller particles with their centrifuge state (by a factor of 4) than a previous method if ref. 41. Moreover, it was shown that with 8 minutes of electrospray deposition, antimicrobial activity was maximized without exhibiting cytotoxicity while also exhibiting the osteogenic activity of the TNTs based on cell morphology. Thus, the authors have convincingly demonstrated that TNT surfaces can be coated with tetracycline nanoparticles from electrospray deposition to produce a surface that is osteogenic and antimicrobial, which has important implications on reducing failure rates in dental implant surgery. This is a significant result and this paper should be accepted.  There are two very minor points the authors should address below.

  1. line 212.  The bacterial count is said to be 25% higher, but the base is 240 CFU/mL in the control and 320 CFU/mL in the T0 group, so this is an increase of 80, which divided by 240 is actually 33% higher!
  2. line 261. Considering the broad audience of nanomaterials (some of whom will be engineers) it is probably necessary to write out that ECM stands for "extracellular matrix"--this is well known to those in dental medicine and engineering, but may be perplexing to some other readers.

Author Response

Reviewer #2

General Comments

This is an extremely well-written manuscript detailing a procedure for depositing the anti-bacterial compound tetracycline onto titania nanotube (TNT) surfaces. Anodized TNT dental implants are used because of their ability to be osseointegrated, yet there has remained some concern in the field because of the potential for bacterial adhesion to the surface and hence bacterial infection at the implant site. Tetracycline in PLGA is deposited by electrospray for various lengths of time to the TNT. The authors demonstrate that they produce smaller particles with their centrifuge state (by a factor of 4) than a previous method if ref. 41. Moreover, it was shown that with 8 minutes of electrospray deposition, antimicrobial activity was maximized without exhibiting cytotoxicity while also exhibiting the osteogenic activity of the TNTs based on cell morphology. Thus, the authors have convincingly demonstrated that TNT surfaces can be coated with tetracycline nanoparticles from electrospray deposition to produce a surface that is osteogenic and antimicrobial, which has important implications on reducing failure rates in dental implant surgery. This is a significant result and this paper should be accepted. There are two very minor points the authors should address below.

Response to General Comments

Thank you for your valuable comments and appreciation for the research work.

Comment #1

(Line 212)

The bacterial count is said to be 25% higher, but the base is 240 CFU/mL in the control and 320 CFU/mL in the T0 group, so this is an increase of 80, which divided by 240 is actually 33% higher!

Response

Thank you for pointing out this error. We have replaced the number “25%” with “33%” to use precise value.

Comment #2

(Line 261)

Considering the broad audience of nanomaterials (some of whom will be engineers) it is probably necessary to write out that ECM stands for "extracellular matrix"--this is well known to those in dental medicine and engineering, but may be perplexing to some other readers.

Response

As per the thoughtful suggestion, the abbreviation “ECM” has been modified to “extracellular matrix”.

Round 2

Reviewer 1 Report

The manuscript still contains some inaccuracies. For example, lines 55-56: “Tetracycline (TC) is a broad-spectrum antibiotic that inhibits bacterial protein synthesis,  periodontitis and osteomyelitis”: “inhibis bacterial protein synthesis” refers to the mechanism of action of TC, while “periodontitis and osteomyelitis” are two diseases…

The most important point, however, is the overall significance of the work which remains low.